# Emerging Lysosomal Functions for Photoreceptor Cell Homeostasis and Survival

**DOI:** 10.3390/cells11010060

**Published:** 2021-12-26

**Authors:** Manuela Santo, Ivan Conte

**Affiliations:** 1Department of Biology, University of Naples Federico II, Strada Vicinale Cupa Cinthia, 26, 80126 Naples, Italy; manuela.santo@unina.it; 2Telethon Institute of Genetics and Medicine, Via Campi Flegrei 34, 80078 Pozzuoli, Italy

**Keywords:** autophagy, photoreceptors, lysosome, retinal degeneration, membrane trafficking

## Abstract

Lysosomes are membrane-bound cell organelles that respond to nutrient changes and are implicated in cell homeostasis and clearance mechanisms, allowing effective adaptation to specific cellular needs. The relevance of the lysosome has been elucidated in a number of different contexts. Of these, the retina represents an interesting scenario to appreciate the various functions of this organelle in both physiological and pathological conditions. Growing evidence suggests a role for lysosome-related mechanisms in retinal degeneration. Abnormal lysosomal activation or inhibition has dramatic consequences on photoreceptor cell homeostasis and impacts extensive cellular function, which in turn affects vision. Based on these findings, a series of therapeutic methods targeting lysosomal processes could offer treatment for blindness conditions. Here, we review the recent findings on membrane trafficking, subcellular organization, mechanisms by which lysosome/autophagy pathway impairment affects photoreceptor cell homeostasis and the recent advances on developing efficient lysosomal-based therapies for retinal disorders.

## 1. Introduction

Lysosomes, previously considered simple static organelles employed in the recycling of cellular waste, are acquiring relevance as dynamic organelles that control cell homeostasis and metabolism in response to environmental cues. They are actively involved in the balance between synthesis and degradation processes and, as such, they are now considered major players in ensuring cell survival.

Their size, number and content vary across cell types. These membrane-bound organelles are characterized by more than 60 acid hydrolases responsible for the degradation and recycling of biological macromolecules. These include extracellular and cellular components delivered to the cytoplasm through endocytosis, phagocytosis and autophagy processes [1,2]. Furthermore, lysosomes have also been associated with secretory pathways, cell signalling, gene regulation, cell adhesion, migration and membrane contact sites [3,4]. The impact of lysosomal dysfunction on cellular health may result from abnormal functioning of lysosomal proteins. Lysosomal dysfunction underpins the pathogenesis of many common genetic and acquired disorders, including neurodegenerative and metabolic diseases, as well as cancer. However, the affected tissue and disease phenotypes are often disease specific. In this review, we focus on the molecular pathways and mechanisms by which lysosomal functions are regulated in photoreceptor cells. These mechanisms may be unique to photoreceptors and represent potential new therapeutic targets to treat retinal diseases.

## 2. Photoreceptor Cells

Photoreceptors are specialized light-sensitive retinal cells involved in converting light stimuli into neural signals for image processing in the brain [5,6]. Rod cells operate under dim lighting conditions and are highly light sensitive, whereas cones exhibit less light sensitivity and are responsible for colour vision and high visual acuity [7,8,9,10]. Rods and cones have five primary structural/functional regions: the outer segment (OS), connecting cilium (CC), inner segment (IS), cell body (CB) and synaptic regions. The OS comprises tightly packed membrane discs containing the visual pigments and other proteins involved in the conversion of light stimuli into electrical signals in a process known as phototransduction (Figure 1 left). The CC allows the trafficking of molecules from the IS to the OS. The IS contains all the organelles including mitochondria, endoplasmic reticulum, Golgi complex and lysosomes. The CB is continuous with the IS and includes the nucleus. Photoreceptor cells terminate with the synaptic region, which is responsible for glutamate-mediated signalling to bipolar cells or other secondary retinal neurons. To avoid light-dependent metabolic stress, and to constantly ensure maximum photosensitivity, photoreceptor cells undergo daily shedding of the most distal portion of the OS [11,12]. Older discs undergo phagocytosis by the adjacent retinal pigment epithelium (RPE), and newly formed discs are continuously incorporated at the base of OS. This is a highly demanding procedure, necessary for photoreceptors’ health and survival, relying on a perfect balance between metabolic and catabolic mechanisms in the IS. The equilibrium between synthesis and degradation is essential to ensure photoreceptor homeostasis and function. Unsurprisingly, lysosomes appear to be ideal candidates to orchestrate the special metabolic needs of these cells.

## 3. Endolysosomal System and Membrane Trafficking in Photoreceptors

Among the different cellular mechanisms essential for proper cellular homeostasis, endocytosis and trafficking systems strongly rely on lysosomes [13,14,15,16]. Much is now known about the molecular mechanisms involved in the endolysosomal system, however, we know less about this process in the retinal photoreceptor cells. Under physiological conditions, cells can internalize the plasma membrane, transmembrane proteins and soluble molecules destined for degradation or recycling through endocytosis. This process begins with the embedding of the plasma membrane, resulting in early endosomes’ formation; this structure passes through a series of maturation events, including morphological changes and pH acidification, ultimately leading to the late endosome stages. Late endosomes undergo homotypic fusions with each other, as well as fusion with lysosomes for cargo degradation. Photoreceptor cells are unique in their highly polarized compartmentalization of cell body/soma and processes including OS, IS and synaptic regions. This organization enables them to form complicated networks that mediate specialized functions. Consistent with this structural complexity, it is also very difficult to characterize the different endolysosomal intermediates at the molecular level. Studies from different laboratories recognized Rab5 as an early endosome marker, while late-endosome/lysosome distinction is hardly detectable, apart from the lack of M6P receptor in the latter [17,18]. Photoreceptors strongly rely on the endolysosomal system for the degradation of rhodopsin which, similarly to other G-protein coupled receptors, is internalized after activation. In healthy photoreceptors, this process is triggered by the transient binding of rhodopsin and arrestin 2 (Arr2), which interact with the AP-2 adaptor protein to enter the endocytic network [19] (Figure 1 right). Taking advantage of different Drosophila mutants, it has been demonstrated that post-translational modifications, of both rhodopsin and arrestin, are crucial in the receptor’s internalization process. In norpA mutants, for example, mutations in the eye-specific phospholipase C result in a block in light-triggered Ca^2+^ dependent phosphorylation of Arr2. Arr2 phosphorylation is essential to release its binding with rhodopsin; as a result, norpA mutants show stable rhodopsin–arrestin complexes which are massively internalized and engulf the endolysosomal system [20,21,22,23]. Similarly, rdgC Drosophila mutants, a loss-of-function model for the Ca^2+^ -dependent serine/threonine protein phosphatase, also show light-dependent photoreceptor degeneration [24]. In this model, the persistence of the phosphorylated form of rhodopsin facilitates the formation of stable rhodopsin–arrestin complexes which accumulate in the internal cellular compartment and trigger apoptosis [25]. Similar retinal degeneration phenotypes are also observed in Drosophila mutants where only the final endosome to lysosome trafficking (*car* mutants) is impaired [26]. Interestingly, the degeneration phenotypes of these flies can be rescued if a mutant form of rhodopsin, lacking the C-terminal phosphorylation sites, is introduced in the *car* background [23]. Altogether, these results strongly support the hypothesis that massive rhodopsin endocytosis can be causative of retinal degeneration observed in flies with both normal or impaired endolysosomal systems. This raises the intriguing question about how photoreceptors regulate the transport and degradation of rhodopsin across their cellular compartments, and how impairment of this process impacts their homeostasis and survival. Interestingly, Hargrove-Grimes et al. identify Rabgef1 as a key factor participating in the photoreceptor’s endolysosomal system by regulating Rab5-GTPase. Rabgef1-KO mice showed reduced early endosome levels and increased autophagic substrates, leading to early loss of both photoreceptor morphology and function [27]. These findings suggest that distal–proximal membrane trafficking can affect photoreceptor homeostasis as endosomes are transported toward the IS, and that they progressively acidify and gain lysosomal hydrolases as they approach the IS. Consistent with this intriguing hypothesis, sphingosine kinases and their metabolites were shown to affect trafficking of the G protein-coupled receptor rhodopsin and the light-sensitive transient receptor potential (TRP) channel by influencing endolysosomal trafficking and altering photoreceptors’ homeostasis [28]. Moreover, endolysosomal system and synaptic vesicle trafficking abnormalities were observed in a knockout zebrafish model for the synaptojanin 1(SynJ1) gene, which was previously demonstrated to participate in the hydrolysis of phosphatidylinositol 4,5-bisphosphate (PI(4,5)P2) and the uncoating of clathrin-coated vesicles [29]. In vivo studies revealed that ablation of SynJ1 induced an aberrant distribution and accumulation of acidic vesicles, aberrant shaped Rab7 positive late endosomes and LC3 positive autophagosomes in the IS, accompanied by late disruption of the Golgi apparatus. Together, these results strongly support the idea that perfect control of endolysosomal system and membrane trafficking events across photoreceptor compartments is necessary to maintain their homeostasis. Consistently, alterations in the morphology and positioning of Arl8, Rab7 and Atg8-carrying endolysosomal and autophagosomal compartments were observed in pre-degenerative conditions in the crumbs mutant Drosophila retina [30].

## 4. Autophagy Pathway

Autophagy is one of the most important biological processes in which lysosomes are involved. It consists of a series of catabolic events essential to ensure the specific degradation of different cellular components (i.e., mitochondria, ribosomes, pathogens, aggregated proteins) to maintain cellular homeostasis and to recycle useful organic substrates [31]. The classical form of autophagy is called macroautophagy and, in an oversimplified view, it can be described as a three-step process starting with (1) formation of the autophagosome, a double-membrane vesicle surrounding degradable cargo, (2) fusion of the autophagosome with a lysosome leading to the formation of an autolysosome and (3) degradation of cargo via acidic hydrolases. Different molecular players are involved in the fine regulation of this process in response to specific cellular needs. According to the specific environmental context, autophagy can function as an adaptive mechanism to avoid cell death under stress conditions, but can also contribute to apoptotic programs if the autophagic stress threshold is exceeded [32]. The pro-survival role of autophagy is represented by its ability to eliminate protein aggregates and non-functional organelles that may be toxic for the cells. Conversely, cytotoxic-autophagy can result from the activation of signals triggering pro-apoptotic pathways (type I cell death) or due to the detrimental effect of massive autophagy (type II cell death). Interestingly, pro-survival and pro-apoptotic signalling machinery can be triggered by the same cue and also share many molecular components, resulting in an intricate network of players. Increased ROS and free calcium concentration, for example, represent two common signals for both activation of apoptotic cell death and pro-survival autophagy pathways [33,34]. The intensity and persistence of the triggering stimulus, as well as the existence of cross-inhibitory loops, will determine the resulting dominant pathways and the final outcome on cell survival [32]. In this context, an important player in regulating the induction of autophagic flux, according to environmental conditions such as nutrient availability, is the AMP-activated protein kinase (AMPK). In particular, upon starvation, AMPK can directly phosphorylate UNC51-like kinase (ULK1) at Ser 317 and Ser 777, allowing the formation of the autophagosome initiation complex together with autophagy-related 13 (ATG13) protein and family-interacting protein of 200 KDa (FIP200) [35]; on the other hand, when nutrients are available, AMPK is inactive and the mammalian target of rapamycin complex 1 (mTORC1) can phosphorylate ULK1 in Ser 757, preventing the interaction between ULK1 and AMPK [35]. AMPK works as an energy sensor to adjust cellular metabolism according to environmental conditions and, by triggering autophagosome formation, is the major regulator of starvation and stress-mediated cell response in different cell types, including photoreceptors (Figure 2). Another player involved in the regulation of AMPK/mTORC1-mediated phosphorylation of ULK1 is the TSC1/2 complex. TSC2, in fact, negatively regulates mTORC1, facilitating pro-autophagic AMPK signalling. Interestingly, it has been demonstrated that, specifically in neurons, TSC2 knock-down, despite being associated with mTORC1 activation, also induces a concomitant increase in AMPK activity, Ser 317/Ser 777 ULK1 phosphorylation and autophagic flux [36]. First, this result reveals that AMPK- and mTORC1- pathways cannot be fully reciprocally inhibited; it further suggests that a low ULK1 phosphorylation level, at Ser 317/Ser 777 residues, might be sufficient to activate autophagic flux. These findings may represent potential safety mechanisms to face neuronal sensitivity to starvation. AMPK activation upon stress condition is a typical trigger of the pro-survival role of autophagy. When nutrients are not available, in fact, autophagy activation is the first compensatory mechanism to provide organic substrates and sustain cell metabolism. Furthermore, it has been demonstrated that AMPK activation is also accompanied by phosphorylation of cyclin-dependent kinase inhibitor p27 which, once stabilized, permits cells to survive starvation by implementing autophagy instead of undergoing apoptosis [37]. In other environmental contexts, however, activation of AMPK signalling can also converge on autophagic-cell death. This aspect, for example, has been addressed in different studies on anticancer compounds which can induce cell death through AMPK-dependent autophagy activation [38,39].

## 5. Autophagosome to Autolysosome Route Impacts on Photoreceptors’ Survival

Once pro-autophagic signals are activated, autophagic related genes (ATGs) and their respective protein products oversee autophagosome formation and maturation [40]. The ATG protein group includes different complexes with specialized functions, such as the Atg5/ULK kinase complex, which recruits Atg13 and FIP200 to promote autophagosome induction [41]; the PI3K3C complex (composed of Atg6/Beclin1, Atg14/ Atg14L, PI3K3C and UVRAG), which recruits other ATG proteins thus contributing to autophagosome formation; and the Atg8/LC3 complex, which is essential for autophagosome closure [42,43]. Different experimental evidence revealed that the basal molecular machinery mediating autophagosome processing is highly conserved among species and is also shared between different cellular subtypes [44]. Consistent with that, the evaluation of the LC3I to LC3-II (lipidated form of the protein) transition is a well-established read-out of autophagy activation and is used to properly interpret autophagic flux, since the total amount of LC3 is not sufficient to allow precise prediction [45]. ATG proteins are significantly involved in the photoreceptor stress-dependent response. For example, knockdown of Atg5 or Beclin in 661W cells is sufficient to rescue cell death induced by hydrogen peroxide treatment, suggesting a triggering role for autophagy in caspase-3 dependent cell death [46]. In vivo studies on Drosophila, however, show that knock-down of or mutation in ATGs negatively impact photoreceptors’ survival due to the accumulation of activated rhodopsin [47]. In line with previous results revealing autophagy-like processes in maintaining a stable level of rhodopsin in photoreceptors [48], the latter study confirms that basal autophagic activity is essential for the normal metabolic requirements of photoreceptor cells. Autophagy activation, by contrast, despite being protective in short-term stress models, can be detrimental for overall retina health (Figure 3). This hypothesis is supported by strong evidence from studies on both murine and human in vitro retinal models of metabolic stress. In particular, short treatment (2 h) of ARPE-19 cells with blockers or inductors of autophagy can increase and reduce cell death mediated by inhibitors of Na^+^/H^+^ exchangers, respectively; however, slightly prolonged treatment (4 h) inverted the result [49]. Similarly, 661W cell death, observed upon prolonged exposure to tamoxifen (18 h), can be prevented via administration of either 3MA, which blocks autophagy activation, or bafilomycin, which blocks autolysosome formation [50]. In vivo studies on Drosophila further confirm the relevance of correct lysosomal fusion events for photoreceptor survival. In flies, the carnation (*car*) gene encodes for the Vps33a protein, which is involved in membrane fusion events. Conditional knock-out of *car* in Drosophila eyes impedes autolysosome formation and leads to autophagosome accumulation, causing light-independent degeneration of photoreceptor cells. These results confirm that the fusion between the autophagosome, carrying the degradative cargo, and the lysosome, providing soluble hydrolases, is also a crucial step impacting the overall success of the degradative machinery in photoreceptors [51].

## 6. Pro-Survival Role of Autophagy in Photoreceptor Cells

In photoreceptors, apart from starvation, autophagy is also triggered by oxidative stress due to light exposure and represents one of the major mechanisms used by these cells to face reactive oxygen species (ROS) damage. In particular, significant insights have been obtained by studying circadian- and light-mediated activation of autophagy pathways in the photoreceptor cells. Firstly, Reme et al. [52] documented a daytime peak of autophagy in the rod photoreceptor IS, characterized by the formation of autophagosomes three hours after the peak period of disc-shedding. Autophagy is, in part, regulated in a circadian manner, but it could be evoked by light pulses in a non-circadian rhythm as measured by conversion of LC3-I to LC3-II in response to light stimuli and oxidative stress [52,53]. However, whether autophagy activation has a positive or negative impact on photoreceptor cells’ survival is a significant and challenging question in the retina-research field [54]. What is becoming undeniable is that the “pros and cons balance” of autophagy induction strongly depends on the precise environmental context in which photoreceptor cells lie. The protective effect of autophagy activation on photoreceptors has been demonstrated in different models of retinal degeneration. Besirli et al. demonstrated that autophagy inhibitor administration is accompanied by an increase in apoptotic photoreceptor cells in in vitro models of retinal detachment [55]. Consistent with that, administration of the autophagy activator rapamycin rescues photoreceptor cell death induced by constant illumination [46]. Similarly, artificial activation of AMPK by the drug metformin protects photoreceptor cells in case of light damage, as demonstrated in the Rd10 model of retinitis pigmentosa (RP) and oxidative stress-induced injury [56]. Despite metformin being reported to significantly induce autophagy [57,58], the positive effect of the drug on photoreceptors is also associated with a general increase in the metabolic activity of these cells, represented by an increase in ATP production, mitochondrial DNA copy number and NADH/NAD^+^ ratio [56]. These results suggest that alternative AMPK pathways, independent from autophagy, might also be activated in photoreceptors to counteract the degeneration associated with aging or oxidative stress. Different experimental evidence, by contrast, supports the hypothesis that mTOR signalling could be a crucial mediator of photoreceptor degeneration in response to nutrient changes in different models of retinitis pigmentosa (RP), including Rd10, Pde6b^−/−^, Pde6g^−/−^, Rho^−/−^ and RHO-P23H transgenic mouse models [59,60]. In particular, transcriptome, molecular and morphological analyses on the retina from these mice revealed that the insulin-mTOR pathway is reduced compared to the WT retina, a condition likely mimicking a cellular starvation signal. Notably, inhibition of mTOR activity via in vivo administration of rapamycin is sufficient to recapitulate cones’ degeneration observed in the different mouse models of RP, and this phenotype can be rescued by systemic treatment of insulin [60]. In all four models of RP, the molecular machinery underpinning mTOR inhibition-dependent cones’ degeneration includes a particular form of autophagy, namely, chaperone-mediated autophagy (CMA), as evidenced by cone-specific increased levels of LAMP2A, a common marker of CMA typically activated upon a prolonged period of starvation [61]. The relevance of mTORC1 activation in cones’ survival has also been demonstrated through cone-conditional knock-out of TSC1 in the Rd1 mouse model of RP, which is sufficient to rescue photoreceptor degeneration by implementing cell metabolism [59]. Consistently, more recently, the same results were further confirmed by in vivo evidence that conditional cone depletion of phosphatase and tensin homologue (pTEN)*,* which activates the PI3K-mTORC1 pathway, is sufficient to recover the retinal phenotype of Rd10 RP mouse model [62]. The apparent discrepancy between the positive effect of both AMPK and mTORC1 signalling on photoreceptors’ health has recently been addressed in a *N*-methyl-*N*-nitrosourea (MNU)-induced photoreceptor cell damage in vitro model [63]. Suppression of the PI3K/mTOR pathway is the major molecular feature of this treatment and is accompanied by autophagy activation and disruption of lysosomal degradation efficiency, ultimately leading to caspase-dependent cell death. Autophagy activation is essential as a primary compensatory mechanism to face MNU-dependent cell stress, as confirmed by aggravated apoptotic cell death upon treatment with autophagy inhibitors. However, the activation of degradative machinery alone is not sufficient to ensure cell survival. The authors, in fact, suggest that impaired lysosomal degradative efficiency might depend on sustained mTORC1 inhibition, which interferes with lysosomal relocation and autolysosome formation [64,65]. Taken together, these results confirm that nutritional imbalance, independent of precise genetic background of the retinal pathology, can be a common feature of photoreceptor degeneration and can be potentially mediated by a non-canonical form of autophagy. Furthermore, these results highlight the mTORC1 pathway as a great candidate for new potential therapeutic approaches in the field of retina degeneration.

## 7. Lysosomal Dysfunction in Retinal Disease

Retinal diseases are a very heterogeneous class of pathologies affecting millions of people worldwide and, as such, represent a major health issue. Despite different molecular backgrounds and specific temporal progression, most retinal diseases are characterized by photoreceptor cells’ loss [66]. In this background, the retinal research field is working on identifying common cellular mechanisms at the base of photoreceptor cell death to be used as hypothetical targets to treat retinal pathologies. Lysosomal and autophagic pathways seem to be potential candidates that are often compromised in different models of retina degeneration and implicated in molecular mechanisms leading to photoreceptor death. However, whether induction of autophagy would result in amelioration or worsening of retinal disease progression is still an open issue, as it strongly depends on the precise pathological context. Recently, Intartaglia et al. reviewed the correlation between defects in lysosomal proteins or lysosomal-related proteins and retinal dystrophies associated with lysosomal storage disorders (LSDs) [67]. Importantly, several lines of study also demonstrated the role of autophagy in RP. RP is a highly heterogeneous inherited retinal dystrophy, characterized by mutations in genes involved in photo-transduction, trafficking and recycling pathways. Its identifying feature is the progressive degeneration and death of rod photoreceptors, followed by non-autonomous death of cones [68,69]. Consistent with that, first symptoms include initial loss of night vision and later onset central vision loss [70]. Experimental evidence revealed that increased autophagic flux can be either protective or deleterious for photoreceptor survival in RP. For example, Yao et al. demonstrated that pharmacological inhibition of autophagy, as well as a rod-specific block of autophagic flux, significantly improved photoreceptor survival in RHO^P23H^ mouse models of RP [71]. Conversely, Rodríguez-Muela et al. showed that the Rd10 mouse retina is characterized by a marked reduction in autophagy flux, calcium overload and calpain activation, resulting in permeabilization of the lysosomal membrane and consequent photoreceptor cell death [72]. Interestingly, induction of autophagy, both in vivo and in vitro, worsens the photoreceptor death phenotype of Rd10 mice instead of rescuing it. Together, these data first confirm that despite resembling a major feature of RP, the precise genetic background of each animal model severely impacts the molecular feature of disease progression [66,73,74]; second, these results suggest that activation of autophagy in photoreceptors can be detrimental in the presence of non-functional lysosome [72].

Age-related macular degeneration (AMD) is another great example of the potential involvement of lysosomes and autophagy in the manifestation of retinal dystrophies [75]. The pathology is characterized by a severe decrease in fine vision due to progressive degeneration of the macula. A major feature of AMD is represented by the accumulation of drusen (deposit of biological material) throughout the retina, which can be nicely recapitulated in vitro and significantly depends on lysosomal malfunction [76]. It has been demonstrated that aging is strongly correlated with the progression of AMD, in fact, molecular machinery essential for degradation of organic substrates and recycling of cellular components loses efficiency with aging and might contribute to the formation of drusen [77,78]. In this context, a major role is played by the autophagic system of RPE which is essential for maintaining healthy photoreceptors [79,80,81] and, as such, inductors of autophagy are tempting therapeutic approach in the treatment of AMD. Interestingly, in aging retinas, a shift from classical autophagy to chaperone-mediated autophagy (CMA) is detectable [82]. Transcriptome and protein levels of Beclin1 and Atg7, two specific regulatory components of the autophagy initiation complex, in fact, are significantly reduced in aged retinal samples. Consistent with this, an up-regulation of LAMP-2A CMA marker is also detectable. According to this study [82], the switch from autophagy to CMA is unidirectional and not reversible with time. If, on one side, the block of autophagy induces activation of CMA as a compensatory mechanism, the inhibition of the latter is not sufficient to re-activate the autophagic machinery. Interestingly, among other retinal cell types, this unidirectional signalling is a unique feature of photoreceptors [82]. At the cellular level, the autophagy to CMA switch is followed by a later lipofuscin accumulation, reduction in the number of photoreceptors in the ONL, outer segment alterations, and an increased number of apoptotic cells [82]. By contrast, an increase in lysosomal-dependent proteolysis was still detectable as a result of CMA upregulation in aged retinas. These results first support the hypothesis that the age-dependent block of the autophagy system takes place at the level of autophagosome formation rather than at the lysosomal-fusion stage; furthermore, they reveal that the physiological activation of CMA is a good compensatory mechanism to significantly delay the macroscopic manifestation of photoreceptors death due to the age-dependent autophagy block [82]. Which upstream signal determines the age-dependent autophagy block and whether this turning point could be important for the treatment of AMD are still unsolved questions. So far, different drugs have been tested in clinical trials, including antioxidants, inhibitors of complement cascades and autophagy triggers, among others [78]. However, given the complexity and heterogeneity of mechanisms leading to retinal degeneration, none of the presently available therapies are sufficient to cure AMD.

## 8. Open Issues

Recent discoveries have demonstrated that lysosomes are dynamically employed in the endolysosomal system, membrane trafficking and circadian and light-induced autophagy pathways to allow photoreceptor cells to adapt to environmental cues. In recent years, growing evidence has shown how numerous molecular pathways controlling the onset and progression of these mechanisms are crucial for the effective recycling of rhodopsin and/or oxidate proteins, both of which are critical to photoreceptors’ homeostasis. Importantly, additional details need to be explored and added to better define molecular mechanisms controlling lysosomal response and function to light, oxidative stress and cell clearance. So far, pieces of evidence from different studies strongly support the hypothesis that lysosome-associated pathways might be powerful candidates to counteract retinal degeneration. In particular, despite being characterized by specific genetic backgrounds, different retinal pathologies share common features including metabolic unbalance, susceptibility to oxidative stress, accumulation of proteins and damaged organelles, which can all be potentially targeted by autophagy modulation. In this respect, the proper understanding of the specific context in which autophagy activation or inhibition could exert a beneficial effect on photoreceptors survival is of extreme importance.

Several questions remain: How does light induce the autophagy pathway in photoreceptor cells? Which signals are required to mediate membrane trafficking across the compartments of photoreceptor cells? Are there specific molecular networks inducing lysosomal involvement in the endolysosomal rather than autophagolysosomal systems? Are there lysosomal proteins contributing to the formation of selective, highly specialized lysosomes to remove phagocyted photo-oxidate rhodopsin? Defects in both the endolysosomal system and membrane trafficking represent a pathogenic cause for retinal degeneration. As knowledge of the functions of lysosomes in photoreceptors’ cell death, homeostasis and function increases, we can look forward to developing new and more promising therapeutic interventions for retinal diseases.

## Figures and Tables

**Figure 1 cells-11-00060-f001:**
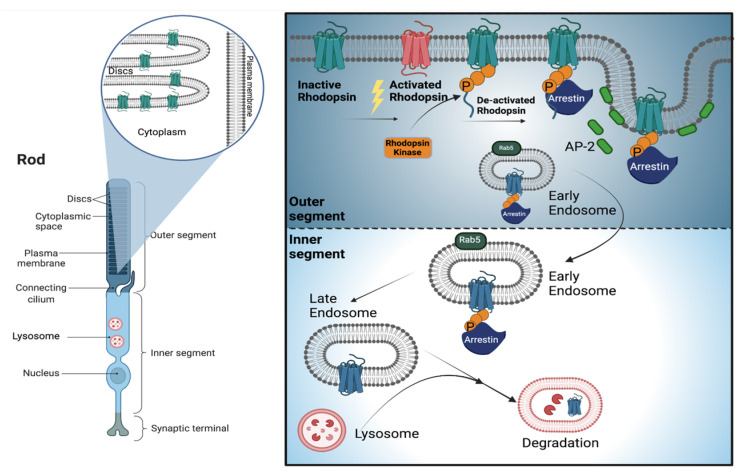
Distal-to-proximal trafficking of rhodopsin in photoreceptors. (**Left**) Schematic representation of primary structural/functional region of rod photoreceptor with a magnification of outer segment discs. (**Right**) After light-induced activation, rhodopsin receptor is de-activated by rhodopsin kinase-dependent phosphorylation and binding with arrestin. The subsequent interaction with AP-2 protein allows the formation of the early endosome which travels from the outer to inner segment to enter the endolysosomal system. Late endosome–lysosome fusion ultimately led to rhodopsin degradation.

**Figure 2 cells-11-00060-f002:**
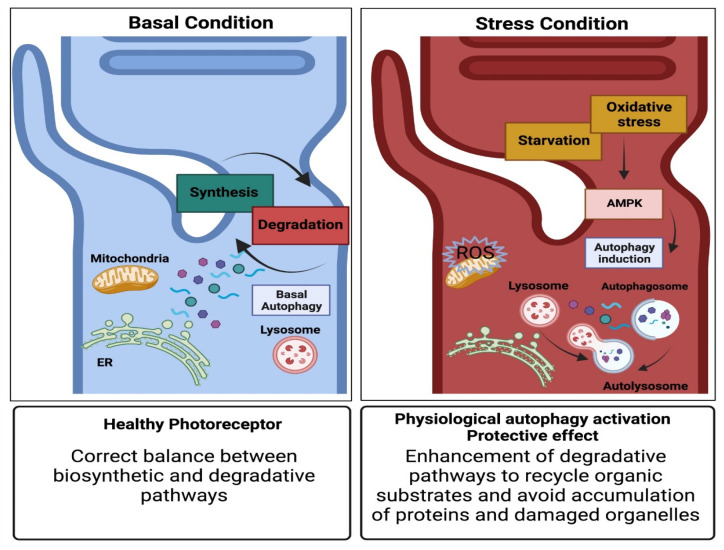
AMPK-dependent stress response in photoreceptors. (**Left**) In basal condition, the healthy state of photoreceptors is ensured by the correct balance between biosynthetic and degradative pathways. (**Right**) In case of stress conditions (starvation or oxidative stress), the AMPK pathway is activated allowing autophagosome formation and autophagy induction as protective, compensatory mechanisms to recycle organic substrates and avoid stress-dependent cell damage.

**Figure 3 cells-11-00060-f003:**
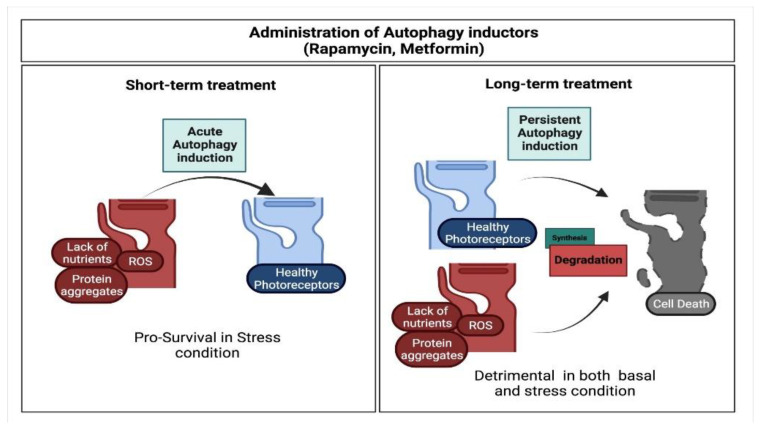
Impact of autophagy activation on photoreceptors’ health. Schematic representation of short- and long-term effects of autophagy activator treatment on photoreceptors’ health.

## Data Availability

Not applicable.

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
