# Peer review of "Emerging Lysosomal Functions for Photoreceptor Cell Homeostasis and Survival"

_cells, 2021, doi:10.3390/cells11010060_

Round 1

Reviewer 1 Report

The manuscript by Santo et al., describes the role of lysosomal function and autophagy on photoreceptor cell survival and degeneration.  The authors have gathered important information related to lysosomes and their involvement in photoreceptor degeneration and retinal diseases. However, there are multiple typos throughout the manuscript and figure legends that need to be corrected.  In addition, the manuscript needs thorough editing as it contains grammatical mistakes with very long sentences that make it difficult to follow.

Also, several references are left out and should be cited, including Mitter et al., 2013, Adv Exp Med Biol; Golestaneh et al., 2017, Cell Death Disease; Patrick M Chen et al., 2011, Cell Biosci.  

Author Response

Reviewer 1

The manuscript by Santo et al., describes the role of lysosomal function and autophagy on photoreceptor cell survival and degeneration.  The authors have gathered important information related to lysosomes and their involvement in photoreceptor degeneration and retinal diseases.

However, there are multiple typos throughout the manuscript and figure legends that need to be corrected.  In addition, the manuscript needs thorough editing as it contains grammatical mistakes with very long sentences that make it difficult to follow.

Answer: We thank the reviewer for his/her time and effort in reviewing our manuscript and for the appreciation of value of our Review. We greatly appreciate the effort the Reviewers went through to improve our manuscript. We also apologize for the presence in the previous version of a number of inaccuracies. We corrected all the typos throughout the manuscript and revised grammatical mistakes. We hope the new version of our manuscript will satisfy the Reviewer’s worries.

Also, several references are left out and should be cited, including Mitter et al., 2013, Adv Exp Med Biol; Golestaneh et al., 2017, Cell Death Disease; Patrick M Chen et al., 2011, Cell Biosci. 

Answer: We thank the referee for these important comments aimed to ameliorate our manuscript. We addressed these concerns, added the refences and modified the manuscript accordingly.

  • Line 426-428 : “Age-related macular degeneration (AMD) is another great example of the potential involvement of lysosomes and autophagy in the manifestation of retinal dystrophies [75].”
  • Line 429-432 : “ A major feature of AMD is represented by the accumulation of drusen (deposit of biological material) throughout the retina which can be nicely recapitulated in-vitro and significantly depends on lysosomal malfunction [76].”
  • Line 435-438 : “In this context, a major role is played by the autophagic system of RPE which is essential for maintaining healthy photoreceptors [79,80,81] and as such, inductors of autophagy are tempting therapeutical approach in the treatment of AMD.”

Reviewer 2 Report

The review article titled “Emerging lysosomal functions for photoreceptor cell homeostasis and survival” by Manuela Santo and Ivan Conte needs further consideration. The comments below highlight some of the questions and suggestions regarding the article.

Comment1: The article needs to be consistent in the terminology and spellings used in text figures as well as quoting references. Several sentences need re-phrasing. Some of them have been noted here.

Abstract: lines 15,16: cellular functions needs for the vision. This could be re-written as, "cellular function needs for vision".

In Line 29, consider restructuring the sentence. In Line 35 dysfunction is misspelled.

Pg2: Use of disk in text is inconsistent with disc in Figure1

Line 85-86. Please revise the sentence.

While quoting studies, in some places the authors were referred to as co-authors (Line 100, line 222) and in other places as et al. (Line 282). In line 292 and 295 the term used was colleagues.

Figure 2 and Figure 3 have several typos (mitochondria, healthy, biosynthetic etc).

Line 173 has some text in different font.

Line 43, 14 124 and 206 the section titles looks orphaned.

Line 280 Please check for spelling, Line 294 needs punctuation.

Figure 2: Oxydative, should be Oxidative; Fig. 2 and 3, Healty, should be corrected to Healthy; biosintetic, should be corrected to Biosynthetic.

Comment 2: Several assertions made in the article need additional references. For example,

Line 47: Additional studies can be referred to underline the role of photoreceptors in vision and role of cones in visual acuity.

Line 73 Please cite additional references that define the role of lysosomes in trafficking and cellular homeostasis.

Line 178 the study cited needs reference.

Line 288 Please cite references for the statement regarding earlier degeneration of rod photoreceptors in RP

Line 319 The statement regarding unidirectional nature of chaperone mediated autophagy needs further supporting references.

Comment 3: In Section 3 line 93, authors need to emphasize how the norpA and rdgc mutations result in dephosphorylation of arrestin2. In these mutants is rhodopsin accumulation subsequent to autophagy signal or does it precede it?

The internalization of rhodopsin is due to rhodopsin-arrestin2 complex formation that is subsequent to the inhibition of phosphorylation in these mutants, not necessarily dephosphorylation of arrestin2?

Comment 4: In section 4, the authors need to make a clear distinction between pro-survival autophagy and pro-apoptotic autophagy in the introduction. This would give further credence to authors’ conclusion in this section that pro-survival and proapoptotic pathways share common signals. This is necessary as the role of kinases like AMPK might vary in both of those conditions.

Comment 5: In section 6, what is the significance of conversion of LC3-I to LC3-II in relation to autophagy? (Line 216)

Additionally, it has been shown that metformin as an AMPK activator, also inhibits mTORC, even in very small concentrations in mouse liver cells. So how might metformin increase the metabolic activity in neurons? (Line 230)

Comment 6: Section 8, needs to summarize the potential of targets in lysosome-mediated autophagy that can be useful in exploring and designing therapeutic approaches in retinal diseases affected by disruption in photoreceptor homeostasis.

Thank you,

Author Response

Reviewer 2

The review article titled Emerging lysosomal functions for photoreceptor cell homeostasis and survival” by Manuela Santo and Ivan Conte needs further consideration. The comments below highlight some of the questions and suggestions regarding the article.

Comment1: The article needs to be consistent in the terminology and spellings used in text figures as well as quoting references. Several sentences need re-phrasing. Some of them have been noted here.

 Answer: We are pleased that the Reviewer found our data 'potentially quite novel and important' and provided us with their insightful comments/suggestions that helped us to improve our manuscript. We are apologized for our grammatic errors, and we have corrected the errors. The text was revised by dr. Phoebe Ashley-Norman, a professional English editor.

Abstract: lines 15,16: cellular functions needs for the vision. This could be re-written as, "cellular function needs for vision”.

Answer: We have now fixed this inaccuracy in the revised manuscript. The text now reads: “Abnormal lysosomal activation or inhibition has dramatic consequences on photoreceptor cell homeostasis and impacts extensive cellular function, which in turn affects vision.” - Line 19-21

In Line 29, consider restructuring the sentence.

Answer: We have changed the text according to his/her suggestion.

  • Line 35-39 : “Their size, number and content vary across cell types. These membrane-bound organelles are characterized by more than 60 acid hydrolases responsible for the degradation and recycling of biological macromolecules. These include extracellular and cellular components delivered to the cytoplasm through endocytosis, phagocytosis and autophagy processes [1,2].”

In Line 35 dysfunction is misspelled.

Answer: We have now fixed this inaccuracy in the revised manuscript

Pg2: Use of disk in text is inconsistent with disc in Figure1

Answer: We unified the use of word “disc” throughout the manuscript, accordingly.

Line 85-86. Please revise the sentence.

Answer: We have changed the text according to his/her suggestion.

- Line 96-99 : “ This organization enables them to form complicated networks that mediate specialized functions. Consistent with this structural complexity, it is also very difficult to characterize the different endolysosomal intermediates at molecular level. ”

While quoting studies, in some places the authors were referred to as co-authors (Line 100, line 222) and in other places as et al. (Line 282). In line 292 and 295 the term used was colleagues.

Answer: We unified the use of “et al.” throughout the manuscript.

Figure 2 and Figure 3 have several typos (mitochondria, healthy, biosynthetic etc).

Answer: We have now fixed this inaccuracy in the revised manuscript

Line 173 has some text in different font.

Answer: We have now fixed this inaccuracy in the revised manuscript

Line 43, 14 124 and 206 the section titles looks orphaned.

Answer: We have now fixed this inaccuracy in the revised manuscript

Line 280 Please check for spelling, Line 294 needs punctuation.

Answer: We have corrected the errors. The text was revised by dr. Phoebe Ashley-Norman, a professional English editor.

Figure 2: Oxydative, should be Oxidative; Fig. 2 and 3, Healty, should be corrected to Healthy; biosintetic, should be corrected to Biosynthetic.

Answer: We have now fixed this inaccuracy in the revised manuscript

Comment 2: Several assertions made in the article need additional references. For example,

 Line 47: Additional studies can be referred to underline the role of photoreceptors in vision and role of cones in visual acuity.

Answer: We cited the following additional studies in lines 51-52:

  1. Lamb, T. D., & Pugh, E. N. Phototransduction, dark adaptation, and rhodopsin regeneration the proctor lecture (2006), Inves- 422 tigative ophthalmology & visual science, 47(12), 5138-5152. 423

  2. Kawamura, S., & Tachibanaki, S. Rod and cone photoreceptors: molecular basis of the di erence in their physiology (2008), 424 Comparative Biochemistry and Physiology Part A: Molecular & Integrative Physiology, 150(4), 369-377. 425

  3. Musta , D., Engel, A. H., & Palczewski, K. Structure of cone photoreceptors (2009), Progress in retinal and eye research, 28(4), 426 289-302. 427

  4. Lamb, T. D. Why rods and cones? (2016), Eye, 30(2), 179-185.

Line 73 Please cite additional references that define the role of lysosomes in trafficking and cellular homeostasis.

Answer: We cited the following additional studies in lines 84-85:

  1. 13. Fernández-Monreal, M., Brown, T. C., Royo, M., & Esteban, J. A. The balance between receptor recycling and tra cking to- 433 ward lysosomes determines synaptic strength during long-term depression (2012), Journal of Neuroscience, 32(38), 13200- 434 13205. 435

  2. Li, X., Garrity, A. G., & Xu, H. Regulation of membrane tra cking by signalling on endosomal and lysosomal membranes 436 (2013), The Journal of physiology, 591(18), 4389-4401. 437

  3. Kim, T., Yamamoto, Y., & Tanaka-Yamamoto, K. Timely regulated sorting from early to late endosomes is required to main- 438 tain cerebellar long-term depression (2017), Nature communications, 8(1), 1-16. 439

  4. Ballabio, A., & Bonifacino, J. S. Lysosomes as dynamic regulators of cell and organismal homeostasis (2020), Nature reviews 440 Molecular cell biology, 21(2), 101-118.

Line 178 the study cited needs reference.

Answer: We cited the following additional studies in lines 264-266  :

  1. Kunchithapautham, K., & Rohrer, B. Apoptosis and autophagy in photoreceptors exposed to oxidative stress (2007),Au- 504 tophagy, 3(5), 433-441.

Line 288 Please cite references for the statement regarding earlier degeneration of rod photoreceptors in RP.

Answer: We cited the following additional studies in lines 408-410

  1. Campochiaro, P. A., & Mir, T. A. The mechanism of cone cell death in Retinitis Pigmentosa (2018), Progress in retinal and eye 551 research, 62, 24-37. 552

  1. Newton, F., & Megaw, R. Mechanisms of photoreceptor death in retinitis pigmentosa (2020), Genes, 11(10), 1120.

Line 319 The statement regarding unidirectional nature of chaperone mediated autophagy needs further supporting references.

Answer: No other references are available in support of this particular experimental evidence. We rephrase the sentence as follows :

Line 443-446 : “According to this study [82], the switch from autophagy to CMA is unidirectional and not reversible with time.”

Comment 3: In Section 3 line 93, authors need to emphasize how the norpA and rdgc mutations result in dephosphorylation of arrestin2. In these mutants is rhodopsin accumulation subsequent to autophagy signal or does it precede it?

 The internalization of rhodopsin is due to rhodopsin-arrestin2 complex formation that is subsequent to the inhibition of phosphorylation in these mutants, not necessarily dephosphorylation of arrestin2?

Answer: Thanks for the reviewer's constructive suggestions. We have now rewritten a large part of the manuscript and removed these sentences, accordingly to reviewer’s comments and suggestions

Lines 106-127 : “ Taking advantage of different Drosophila mutants, it has been demonstrated that post-translational modifications, of both rhodopsin and arrestin, are crucial in the receptor internalization process. In norpA mutants, for example, mutations in the eye-specific phospholipase C results in a block in light-triggered Ca2+ dependent phosphorylation of Arr2. Arr2 phosphorylation is essential to release its binding with rhodopsin, as result, norpA mutants show stable rhodopsin-arrestin complexes which are massively internalized and engulf the endolysosomal system [20,21,22,23]. Similarly, rdgC Drosophila mutants, a loss-of-function model for the Ca2+ -dependent serine/threonine protein phosphatase, also show light-dependent photoreceptor degeneration [24]. In this model, the persistence of the phosphorylated form of rhodopsin facilitates the formation of stable rhodopsin-arrestin complexes which accumulate in the internal cellular compartment and trigger apoptosis [25]. Similar retinal degeneration phenotypes are also observed in Drosophila mutants where only the final endosome to lysosome trafficking (car mutants) is impaired [26]. Interestingly, the degeneration phenotypes of these flies can be rescued if a mutant form of rhodopsin, lacking the C-terminal phosphorylation sites, is introduced in the car background [23]. Altogether these results strongly support the hypothesis that massive rhodopsin endocytosis can be causative of retinal degeneration observed in flies with both normal or impaired endolysosomal systems.”

Comment 4: In section 4, the authors need to make a clear distinction between pro-survival autophagy and pro-apoptotic autophagy in the introduction. This would give further credence to authors’ conclusion in this section that pro-survival and proapoptotic pathways share common signals. This is necessary as the role of kinases like AMPK might vary in both of those conditions.

Answer: Thanks for the reviewer's constructive suggestions. We have now clarified the distinction between pro-survival autophagy and pro-apoptotic autophagy as follows :

Line 163-168 : “Different molecular players are involved in the fine regulation of this process in response to specific cellular needs. According to the specific environmental context, autophagy can function as an adaptive mechanism to avoid cell death under stress conditions, but can also contribute to apoptotic programs if the autophagic stress threshold is exceeded [32]. The pro-survival role of autophagy is represented by its ability to eliminate protein aggregates and non-functional organelles that may be toxic for the cells. Conversely, cytotoxic-autophagy can result from the activation of signals triggering pro-apoptotic pathways (type I cell death) or due to the detrimental effect of massive autophagy (type II cell death). Interestingly, pro-survival and pro-apoptotic signalling machinery can be triggered by the same cue and also share many molecular components resulting in an intricate network of players. Increased ROS and free Calcium concentration, for example, represent two common signals for both activation of apoptotic cell death and pro-survival autophagy pathways [33,34]. The intensity and persistence of the triggering stimulus, as well as the existence of cross-inhibitory loops, will determine the resulting dominant pathways and the final outcome on cell survival [32]”.

Line 199-209 : “AMPK activation upon stress condition is a typical trigger of the pro-survival role of autophagy. When nutrients are not available, in fact, autophagy activation is the first compensatory mechanism to provide organic substrates and sustain cell metabolism. Furthermore, it has been demonstrated that AMPK activation is also accompanied by phosphorylation of cyclin-dependent kinase inhibitor p27 which, once stabilized, permits cells to survive starvation by implementing autophagy instead of undergoing apoptosis [37]. In other environmental contexts, however, activation of AMPK signalling can also converge on autophagic-cell death. This aspect, for example, has been addressed in different studies on anticancer compounds which can induce cell death through AMPK-dependent autophagy activation [38,39].”

Comment 5: In section 6, what is the significance of conversion of LC3-I to LC3-II in relation to autophagy? (Line 216)

Answer: Thanks for the reviewer's constructive suggestions. We have now expanded the significance of conversion of LC3-I to LC3-II to the section 5 as follows :

- Line 255-261 : “ Different experimental evidence revealed that the basal molecular machinery mediating autophagosome processing are highly conserved among species and are also shared between different cellular subtypes [44]. Consistent with that, the evaluation of the LC3I to LC3-II (lipidated form of the protein) transition, is a well-established read-out of autophagy activation and is used to properly interpret  autophagic flux , since the total amount of LC3 is not sufficient to allow precise prediction [45].”

Additionally, it has been shown that metformin as an AMPK activator, also inhibits mTORC, even in very small concentrations in mouse liver cells. So how might metformin increase the metabolic activity in neurons? (Line 230)

Answer: We do agree with the reviewer and have rewritten part of the text to incorporate the description of the role of metformin in the photoreceptor cells as follows :

-Line 345-349: “Despite metformin being reported to significantly induce autophagy [57, 58], the positive effect of the drug on photoreceptors is also associated with a general increase in metabolic activity of these cells represented by an increase in ATP production, mitochondrial DNA copy number and NADH/NAD+ ratio [56].”

Comment 6: Section 8, needs to summarize the potential of targets in lysosome-mediated autophagy that can be useful in exploring and designing therapeutic approaches in retinal diseases affected by disruption in photoreceptor homeostasis.

Answer: We have now summarized this point and rewritten the sentence accordingly.

-Line 474 - 482 : “ So far, pieces of evidences from different studies strongly support the hypothesis that lysosome-associated pathways might be powerful candidates to counteract retinal degeneration. In particular, despite being characterized by specific genetic backgrounds, different retinal pathologies share common features including metabolic unbalance, susceptibility to oxidative stress, accumulation of proteins and damage organelles which can all be potentially targeted by autophagy modulation. In this respect, the proper understanding of the specific context in which autophagy activation or inhibition could exert a beneficial effect on photoreceptors survival is of extreme importance”.

Round 2

Reviewer 2 Report

Authors have answered all my comments and made revisions accordingly.